# Optimizing the Antimicrobial Activity of Sodium Hypochlorite (NaClO) over Exposure Time for the Control of *Salmonella* spp. In Vitro

**DOI:** 10.3390/antibiotics13010068

**Published:** 2024-01-10

**Authors:** Nathaly Barros Nunes, Jaqueline Oliveira dos Reis, Vinicius Silva Castro, Maxsueli Aparecida Moura Machado, Adelino da Cunha-Neto, Eduardo Eustáquio de Souza Figueiredo

**Affiliations:** 1Faculty of Agronomy and Zootechnics, Federal University of Mato Grosso (UFMT), Cuiabá 78060-900, MT, Brazil; nathalyb.nunes@outlook.com (N.B.N.); jaque10nutri@gmail.com (J.O.d.R.); 2Faculty of Nutrition, Federal University of Mato Grosso (UFMT), Cuiabá 78060-900, MT, Brazil; adeneto40@gmail.com; 3Food Science Program, Federal University of Rio de Janeiro (UFRJ), Rio de Janeiro 21941-630, RJ, Brazil; maxsuelii@hotmail.com

**Keywords:** fish, *Salmonella* Enteritidis, *Salmonella* Schwarzengrund, NaClO, exposure time, water temperature, CCRD

## Abstract

Fish is a nutritionally rich product; however, it is easily contaminated by pathogenic microorganisms, such as *Salmonella* spp. Therefore, this study aimed to identify the best concentration of sodium hypochlorite (NaClO), exposure time, and water temperature that allow the most effective antimicrobial effect on the viable population of *Salmonella* spp. Thus, *Salmonella* Enteritidis ATCC 13076 and *Salmonella* Schwarzengrund were exposed to different time frames, ranging from 5 min to 38.5 min, temperatures between 5 and 38.5 °C, and NaClO concentrations ranging from 0.36 to 6.36 ppm, through a central composite rotational design experiment (CCRD). The results demonstrated that the ATCC strain exhibited a quadratic response to sodium hypochlorite when combined with exposure time, indicating that initial contact would already be sufficient for the compound’s action to inhibit the growth of the mentioned bacteria. However, for *S.* Schwarzengrund (isolated directly from fish cultivated in aquaculture), both NaClO concentration and exposure time significantly influenced inactivation, following a linear pattern. This suggests that increasing the exposure time of NaClO could be an alternative to enhance *Salmonella* elimination rates in fish slaughterhouses. Thus, the analysis indicates that the *Salmonella* spp. strains used in in vitro experiments were sensitive to concentrations equal to or greater than the recommended ones, requiring a longer exposure time combined with the recommended NaClO concentration in the case of isolates from aquaculture.

## 1. Introduction

Aquaculture is considered a zootechnical activity that has been standing out as an economic alternative, both for large, medium, and small production, as it is a way to take advantage of inactive rural areas [1]. It is estimated that Brazil will register a growth of 104% in fisheries and aquaculture production by 2025 [2]. According to *The Fisheries Yearbook 2022*, the State of Mato Grosso has been standing out in fish production, occupying the 7th position among the 10 largest producers of fish from fish farming in Brazil, producing approximately 42,600 tons [3].

Fish and fish products represent an important source of consumption by the population, in addition to being nutritionally rich with a high content of polyunsaturated fatty acids, proteins of high biological value, and easy digestibility [4]. Thus, fish has been highly consumed around the world due to its high nutritional value and potential health benefits [5].

Fish is a highly perishable food, so it requires a lot of care in handling, ranging from the capture process to storage and marketing. These steps are important to maintain the shelf life of the food [2]. The biggest concern is in relation to its food safety, where it is necessary to avoid contamination by pathogenic microorganisms that could harm the health of consumers [6]. Several pathogens can be present in fish, including *Salmonella* spp. [7].

*Salmonella* spp. is not a biological contaminant originally reported in fish, being introduced through contaminated water or improper handling [8]. There are about 2.659 serovars in the species [9], with the Typhimurium serotype being the most prevalent in products of animal origin [10]. In the US, *S.* Enteritidis, *S.* Typhimurium, and *S.* Newport represent the *Salmonella* spp. transmitted by frequently isolated foods [11]. However, in Brazil, there are few epidemiological studies that indicate *Salmonella* Typhimurium in fish [12], although other serotypes such as *S.* Abony and *S.* Schwarzengrund have been found in fish [13]. There is also evidence of the circulation of the *S.* Typhimurium serotype in fish from aquaculture in the State of Mato Grosso [12].

Due to increased consumption of farmed fish, the contribution of fishery products to foodborne disease outbreaks has increased [12]. *Salmonella* spp. has been reported as one of the most important causes of foodborne disease prevalence [14]. Food safety studies conducted by the European Center for Disease Prevention and Control have shown that almost 7% of foodborne disease outbreaks in the European Union are associated with the consumption of fish and fishery products [15]. In Brazil, more than 6000 cases of food-borne outbreaks have been officially reported from 2007 to 2016, of which 94 were caused by fish and seafood [12].

The presence of this pathogen may be related to several factors, such as the high density of fish biomass in a limited area in intensive farming systems, the access of wild and domestic animals in these areas, effluents, leaching, contaminated food, and human interventions. All these factors can lead to the contamination of pathogens such as *Salmonella* spp. [16]. Contamination of the aquatic ecosystem by *Salmonella* spp. makes the environment a source of dissemination of this microorganism, as this bacterium can survive in the environment for long periods [17]. Furthermore, it can be introduced through contact with the feces of other animals and plants [18].

In fish slaughterhouses, control of *Salmonella* spp. is a challenge, and concerning the efficiency of the forms of its control, it is still not well known [12]. In the animal products industry, chlorine and its derivatives are widely used agents to eliminate pathogenic microorganisms [19]. Chlorinated products are used as the main form of microbiological control in the food industry. The chlorine substance contains hypochlorous acid, which is an effective antioxidant with a neutral charge, enabling easy penetration through the outer membrane of bacteria [20].

The maximum concentration recommendation of sodium hypochlorite allowed by the Ministry of Agriculture, Livestock and Food Supply of Brazil (MAPA) is up to 5 parts per million (ppm) of free residual chlorine [21]. In the fish industry, the process of washing fish with sodium hypochlorite can be considered a critical point of risk control, being the main process to control *Salmonella* spp. However, a study carried out in the State of Mato Grosso indicated that even after washing in chlorinated water at the authorized concentration and time, in the evisceration stages and in the pre-packaging area of a refrigerated slaughterhouse, it was possible to isolate *Salmonella* Ndolo, Mbandaka, and Typhimurium [12]. Other studies from around the world have already evaluated chlorine concentrations higher than those authorized by Brazil (5 ppm) and shorter exposure times for controlling *Salmonella* spp., for instance, in the decontamination process of *Pangasius* fish and in the chlorination of water used in poultry cooling systems [20,22].

Therefore, the present study seeks to determine the best point of antimicrobial activity, considering the concentration of sodium hypochlorite recommended by Brazilian authorities [21], as well as concentrations below and above it, associated with the exposure time and the temperature of the water, to enable a significant inactivation of *Salmonella* Enteritidis ATCC 13076 and *Salmonella* Schwarzengrund in vitro.

## 2. Results

### 2.1. Central Composite Rotatable Design (CCRD)

The combination of NaClO concentration, exposure time, and water temperature in relation to *Salmonella* Enteritidis ATCC 13076 and *S.* Schwarzengrund inactivation has shown statistically significant differences for both strains analyzed. For the ATCC strain, the data indicated a quadratic behavior for chlorine and a linear relationship with time. The results for the ATCC standard strain suggested that longer exposure times under the same chlorine concentration may diminish the agent’s effect (Figure 1a).

Also, it is important to note that the concentration of 5 ppm of sodium hypochlorite recommended by Brazilian legislation for 5 min and at a temperature of 5 °C was able to reduce the bacterial load from 8.3 Log_10_ CFU/mL to 3.1 Log_10_ CFU/mL, representing a reduction of 5.8 Log_10_ CFU/mL (point 14). As for the concentration of 5 ppm for 30 min at 5 °C, there was a reduction of 3.5 Log_10_ CFU/mL (point 15). It can also be seen that increasing the concentration of NaClO to 6.36 ppm had a greater influence on the reduction, reducing a population of 8.5 Log_10_ CFU/mL from the initial load to 7.5 Log_10_ CFU/mL (Table 1).

Concentrations of sodium hypochlorite of ≤1 ppm were also tested with exposure times of 5 and 30 min at temperatures of 5 °C, 30 °C, and 17.5 °C (points 1–5), verifying the growth of *Salmonella* Enteritidis ATCC 13076 ≤ 2 Log_10_ CFU/mL (Table 1). The tested concentrations of 3 ppm of NaClO for this strain were maintained with a reduction of ≥2 Log_10_ CFU/mL.

The combination of treatments for the inactivation of *Salmonella* Schwarzengrund is shown in Figure 1b and Table 1. The result found for this strain was similar to that of the *S.* Enteritidis ATCC 13076 strain, showing a significant reduction in bacterial growth at a concentration of 5 ppm of NaClO for 30 min and 5 °C, reducing a population of 8.5 Log_10_ CFU/mL from the initial load to 1.3 Log_10_ CFU/mL, representing a reduction of 7.2 Log_10_ CFU/mL (point 15). Still at the same concentration of NaClO and the exposure time different, adjusting the temperature to 30 °C (point 16), it could be observed through the reduction of 9.7 Log_10_ CFU/mL of the initial load to 3 Log_10_ CFU/mL that the reduction was 6.1 Log_10_ CFU/mL.

At a concentration of 1 ppm of NaClO at times of 5 and 30 min and at temperatures of 5 °C, 17.5 °C, and 30 °C, bacterial reduction was not achieved, maintaining ≤ 4 Log_10_ CFU/mL; however, even so, it showed a reduction of almost 2 Log_10_ CFU/mL more than when the same NaClO concentration, time, and temperature were applied to the *S.* Enteritidis ATCC 13076 strain. At a concentration of 1 ppm of NaClO at times of 5 and 30 min and at temperatures of 5 °C, 17.5 °C, and 30 °C, the treatment was ineffective, maintaining ≤ 4 Log_10_ CFU/mL; however, even so, it showed a reduction of almost 2 Log_10_ CFU/mL more than when the same NaClO concentration, time, and temperature were applied to the *S.* Enteritidis ATCC 13076 strain.

### 2.2. Validation of the Inactivation Model of Salmonella Enteritidis ATCC 13076 and Salmonella Schwarzengrund

The validation of the experimental model using five random points is presented in Table 2. The results for the *S.* Enteritidis ATCC 13076 strain indicate that in points 1 and 3, the observed values were below the values predicted using the model; values are expressed in Log_10_ CFU/mL. However, in points 2, 4, and 5, the predicted values were lower than the observed values. For *S.* Schwarzengrund, we obtained observed values higher than the values predicted using the experimental model at all points.

### 2.3. Performance of the Experimental Model and Data Adjustment

To assess the quality of the fit of the models obtained and the normality of the data as well as the normality of the residuals, the performance indices are presented in Table 3. The results indicate that the data’s normality was considered acceptable, with *p* > 0.05, as was the case for the residuals of the model. The determination coefficient (R^2^adj) was deemed suitable, being >80 for the tested strains. The MSE showed low values, indicating the absence of experimental errors. Bias factor indices (B_f_), calculated by averaging the observed and predicted validation data, showed values close to 1 for both strains. The same was observed for the accuracy factor (A_f_), which showed values near 1, indicating a minimal mean distance and data dispersion.

## 3. Discussion

Chlorine and its various forms are the compounds commonly used for disinfection in the food and food service industries [23]. According to Brazilian legislation, NaClO is one of the chlorinated products authorized for washing fish in the processing stages [21]. The recommended concentration is 5 ppm, as specified by MAPA and adopted by Brazilian slaughterhouses, despite the cases of isolated *Salmonella* strains in the processing stages in fish slaughterhouses in Brazil [12]. The use of a chemical disinfectant agent such as chlorine must be able to reduce pathogenic bacteria like *Salmonella* by at least 5 Log_10_ CFU/mL [24]. Our results support this information, as the use of 5 ppm of NaClO, an exposure time of 5 min, and a water temperature of 5 °C proved to be effective, resulting in the inactivation of 5.8 Log_10_ CFU/mL in *Salmonella* Enteritidis ATCC 13076. However, for this strain, based on statistical analyses, only the NaClO concentration was significantly affected. A significant difference was observed with a *p* value < 0.05, which was not the case for the variables of temperature and exposure time.

One of the factors highlighted [25] is that the effectiveness of chlorine-based disinfectants is influenced by temperature, concentration, and contact time. For the *Salmonella* Schwarzengrund strain, the results were similar to those of *S.* Enteritidis ATCC 13076. At 5 ppm, it was able to inactivate bacterial growth within 30 min at a temperature of 5 °C, reducing the microbial load by more than 7 Log_10_ CFU/mL. However, raising the water temperature to 30 °C with an exposure time of 5 min resulted in a reduction of 6.1 Log_10_ CFU/mL, indicating that higher water temperatures negatively impact the action of NaClO. Thermal resistance mechanisms are directly associated with cellular permeability, involving changes in the lipid composition of the membrane [26].

In a previous study [13], the presence of *Salmonella* Schwarzengrund was identified in fish processing environments that utilize NaClO. In the current study, this strain demonstrated resistance to variations in NaClO concentration. Through statistical analysis, it was determined that *p* < 0.05 for both the NaClO concentration and time variables, indicating that its microbial load did not decrease. The resistance of pathogens to widely used disinfectants in food companies and industries may contribute to the involvement of specific microorganisms in foodborne outbreaks [23].

Chlorinated compounds do not leave flavors in products as long as they are used in appropriate concentrations [27]. Therefore, a concentration of 6.36 ppm, higher than the recommended level, was chosen to test the hypothesis that a greater concentration results in more effective inactivation of the tested strains. Precisely because this concentration exceeds the limit, it becomes impractical for industrial use. However, our results showed a greater reduction in the microbial load at this concentration; both strains studied were reduced by more than 7 Log_10_ CFU/mL. It is expected that the higher the concentration of NaClO, the greater the inactivation of the microorganism. The National Health Surveillance Agency (ANVISA) approves the use of sanitizers at a concentration of 5 ppm, as this concentration is sufficient to achieve a satisfactory effect without compromising the authenticity and quality of the food, being the only sanitizer permitted to come into direct contact with the fish [21].

A study [20] demonstrated the inefficiency of chlorine used in the quality control of Pangafish (*Pangasius hypophthalmus*) in the fish industry. This was observed through the application of 10, 20, 50, and 120 ppm for durations of 10, 20, and 240 s, with one of the main reasons being the concentration used in the study. Therefore, in this study, we tested varying concentrations of NaClO at varying times and temperatures to optimize the use of this substance in refrigerated slaughterhouses.

Chlorinated compounds do not impart flavors to products, provided they are used in appropriate concentrations [27]. Therefore, the concentration of 6.36 ppm used in our study is unfeasible for industries as it exceeds the recommended level. Although NaClO is a widely used chemical compound, there are no documented reports in the literature about its potential risks to human health. It is important to highlight, however, that the correct use of NaClO is essential in all stages of processing fish and other food preparations, as well as in cleaning processes such as washing the inside of refrigerators. Your inappropriate use of NaClO can lead to changes in the odor or flavor of food.

Adaptations to sanitizers, which can lead to pathogenic bacteria developing resistance, should be avoided by not using them below the recommended concentrations [28]. Prudent use of NaClO is essential to maintaining its effectiveness against microorganisms [29]. We observed that concentrations of 1 ppm, 3 ppm, and 0.36 ppm were not sufficient for effective inactivation, resulting in only a 1 Log_10_ CFU/mL reduction (Table 2).

There is limited information regarding the effect of free chlorine concentrations on bacterial survival in fish [20], highlighting the importance and relevance of this study due to the scarcity of literature on this topic. Adequate NaClO concentration and exposure time are crucial to eliminating pathogens present on the surface of the fish, as reported in previous studies [23,30]. Disinfectant contact time and concentration can significantly affect the effectiveness of the disinfection process.

In the present study, it was verified that the *Salmonella* Enteritidis ATCC 13076 strain was not influenced by the contact time or temperature but by the NaClO concentration, while the *Salmonella* Schwarzengrund strain was influenced by the NaClO concentration and exposure time but not by the temperature. The fish washing step aims to eliminate bacteria from the processing environment, and the mucus present on the surface of the fish, where the glycoproteins released by the skin glands are located, creates a favorable environment for the growth and proliferation of undesirable microorganisms [12]. The correct washing with NaClO is a preventive factor for the control of pathogenic microorganisms in order to avoid problems in the health of the consumer who purchased this product [31].

The survival data of the *Salmonella* Enteritidis ATCC 13076 and *Salmonella* Schwarzengrund strains tested were normal (*p* > 0.05), as well as for the model residues, as shown in Table 2. The adjusted coefficient determination (R^2^adj) for both strains was considered adequate (>80). The mean square error model (MSE) indicated low values, meaning that there were no experimental errors, since MSE values symbolize possible variability and errors in the experiment [32], being able to measure the reliability of the data obtained [26].

Bias factor indices (B_f_) measuring the mean agreement between predicted and observed data [33] were calculated through validation experiments with five random points that were not used in the experimental model, with results close to 1. The precision factor (A_f_) indicates the average distance and dispersion of the data in relation to the measure of equivalence [34]. The expected range for A_f_ in the variables is between 0.10 and 0.15 [35]. In current experimental models, a range of 1.2 to 1.4 is considered acceptable. Our results corroborate these standards in current experimental models. The lack of fit provides an estimate of the fit of the experimental model. The lack of fit provides an estimate of how well the experimental model fits the data, determining whether the model is adequate. In the present study, this was found to be not significant for the models, with a *p*-value greater than 0.05, as expressed in Table 4. Thus, R^2^adj, MSE, B_f_, A_f_, and lack of fit demonstrate that the experimental model was adequate to measure the survival of *S.* Enteritidis ATCC 13076 and *S.* Schwazengrund submitted to selected sodium hypochlorite concentrations during varying exposure times and at different temperatures.

In view of this, the relevance of this study is highlighted with regard to the adoption of effective procedures for the control of *Salmonella* spp. in fish processing environments in refrigerated slaughterhouses, because in the absence of these procedures, fish become an environment conducive to the proliferation of pathogenic microorganisms. For example, more efficient exposure times within industries combined with the use of NaClO concentrations of 5 ppm.

## 4. Materials and Methods

### 4.1. Experimental Design

To carry out this study, strains of *Salmonella* Enteritidis ATCC 13076 and *Salmonella* Schwarzengrund previously isolated [13] and stored at −80 °C were revitalized from the bacteriotheque of the Molecular Microbiology of Food Laboratory—LabMMA. A total of 18 points were performed according to the central composite rotatable design (CCRD), followed by validation of the results with 5 additional randomized experiments, followed by analysis of the results (Figure 2). All experimental steps were carried out at the Federal University of Mato Grosso—UFMT, at the Food Molecular Microbiology Laboratory (LabMMA).

### 4.2. Preparation of the Strains

*Salmonella* Enteritidis ATCC 13076 and *Salmonella* Schwarzengrund, previously isolated from eviscerated and frozen tambaqui (*Colossoma macropomum*) [13], were used in the present study. Both were preserved in brain heart infusion (BHI) broth (KASVI^®^, São José dos Pinhais, Brazil) + 15% (v/m) glycerol (Labsynth^®^, São Paulo, Brazil) and stored at −80 °C.

The strains were revitalized in 9 ml of BHI broth (KASVI^®^, Spain) and subsequently incubated at 37 °C for 24 h. Then, the tubes containing the growing bacteria were centrifuged at 2000 rpm for 10 min. The supernatant was discarded, and the bacterial pellet was resuspended in 9 ml of saline solution (0.85%). The concentration of the initial population (*N*_0_) was quantified before the development of the experiment through surface plating.

### 4.3. Central Composite Rotatable Design (CCRD)

The CCRD was used under an arrangement of factors and variables, conducted at 18 points in order to evaluate the combined effect of sodium hypochlorite concentration (ppm), exposure time (minutes), and temperature (Celsius) on the survival of *Salmonella* spp. in fish. The central combination of the parameters, defined as the central point (3 ppm, 17.5 °C, and 17:50 min), was performed in 4 repetitions to measure possible experimental errors and lack of fit of the model (Table 1).

The quantification of the initial bacterial load was performed with inoculum at 7 log_10_ CFU/mL. For each CCRD experiment, 300 mL of sterile water was used to dilute 12% NaClO (Chloro MT^®^, Cuiabá, Brazil) at the desired concentration and at the target temperature of the experiment. NaClO concentrations were measured using the Exact^®^ Micro7+ equipment (Rio Grande do Sul, Brazil), using the Micro7+/Micro20 free chlorine test strip (AKSO^®^, São José dos Pinhais, Brazil) for testing.

Then, 100 µL of bacterial suspension was added to a 9 mL test tube containing sterile water with the desired NaClO concentration, timing the time of exposure to chlorine. After counting the time, 10% thiosulfate (Labsynth^®^, São Paulo, Brazil) was used to neutralize the action of chlorine, and serial dilutions from 10^−1^ to 10^−5^ were performed, followed by surface plating with 100 µL of bacterial suspension in plates of glass. Petri dishes were coated with nutrient agar (KASVI^®^, São José dos Pinhais, Brazil) and incubated at 37 °C for 24 h. After this period, colonies were counted using an electronic plate counter (Edulab, Curitiba, Brazil).

### 4.4. Model of Inactivation and Validation of Salmonella Enteritidis ATCC 13076 and Salmonella Schwarzengrund

To validate the experimental model, 5 random points were analyzed to compare the values observed and predicted using the model in both strains, where NaClO concentrations and exposure time varied, being 1.5 ppm in 25 min, 2.5 ppm in 8 min, 3 ppm in 19 min, 3.5 ppm in 2 and a half minutes, and 4 ppm in 3 and a half minutes. The temperature was the same for the 5 points, being 17.5 °C. After reaching the time required for each experiment, we neutralized the action of NaClO with 10% thiosulfate (Labsynth^®^, Brazil), and within 24 h, we observed the surviving colonies for counting. The results were analyzed using multiple regression analysis [36].

### 4.5. Performance of the Experimental Model and Data Adjustment

To determine the bacterial survival (*SB*), the initial count (*N*_0_) was subtracted from the final count (*N*) in log (*SB* = *N*_0_ − *N*), and the results obtained through the variables used in this study were analyzed through the response surface methodology.

The data obtained were fitted to a polynomial model for each strain of *Salmonella* spp., in which the statistically significant terms (*p* < 0.05) of the model were maintained. The formality of model adjustment was established through the adjusted coefficient of determination (R^2^adj). The model mean square error (*MSE*) [32], performance bias factor (*B_f_*), and accuracy factor *(A_f_*) indices [34] were calculated using the following equations:$$A_{f}=exp\left(\sqrt{\frac{\sum_{k=1}^{m}{\left(Lnf\left(x^{\left(k\right)}\right)-{Ln\mu }^{\left(k\right)}\right)}^{2}}{m}}\right)$$
$$B_{f}=exp\left(\sqrt{\frac{\sum_{k=1}^{m}\left(Lnf\left(x^{\left(k\right)}\right)-{Ln\mu }^{\left(k\right)}\right)}{m}}\right)$$
$$MSE=\frac{1}{m}\sum_{i=1}^{m}{\left({predicted}_{i}-{observed}_{i}\right)}^{2}$$
where $Ln\;f\left(x\right)$ are the values predicted using the model, Lnμ are the observed values, and m is the number of experiments performed in validation [26].

### 4.6. Data Analysis

The experimental data from the CCRD and all other data obtained were analyzed using the Statistica 10.0 software (StartSoft Power Solutions, Tulsa, OK, USA). Data normality, residuals, and lack of fit were checked using the Shapiro–Wilk test.

## 5. Conclusions

It was observed that the inactivation pattern caused by sodium hypochlorite was quadratic for the ATCC strain and linear for the wild strain (*S.* Schwarzengrund). In this way, the results suggest that the strain originating from the aquaculture system (*S.* Schwarzengrund) may perhaps be combated by increasing the exposure time to NaClO, as even with the prolonged exposure, the elimination showed a linear profile (as both chlorine concentration and exposure time increase, the effectiveness of *Salmonella* elimination also grows).

Furthermore, the isolated strains were evaluated following the model obtained using CCRD, which demonstrated good values in quality controls (A_f_, B_f_, and MSE). Therefore, this study suggests that increasing the fish exposure time to NaClO in the industry (specifically 30 min) could be a promising strategy for combating this pathogen in the final product. It is important to highlight that the standardization developed in this study, which involves using a combination of exposure time and NaClO concentration to reduce populations of contaminating microorganisms (pathogens), was not carried out in vivo (on fish).

## Figures and Tables

**Figure 1 antibiotics-13-00068-f001:**
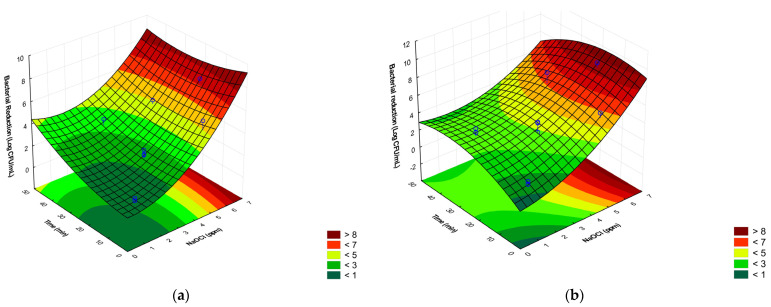
Survival (Log (N_0_ − N) of *Salmonella* Enteritidis ATCC 13076 (**a**) and *Salmonella* Schwarzengrund (**b**) after exposure to sodium hypochlorite, time, and water temperature. Green indicates higher survival after treatments, and red indicates lower survival after treatments.

**Figure 2 antibiotics-13-00068-f002:**
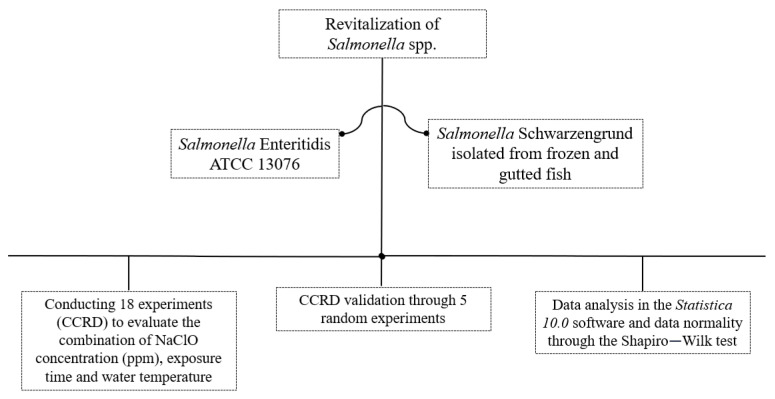
Schematic flowchart of the experimental design.

**Table 1 antibiotics-13-00068-t001:** CCRD array for Salmonella Enteritidis ATCC 13076 and Salmonella Schwarzengrund under sodium hypochlorite concentration, exposure time, and temperature.

Points	Chlorine (ppm)	Time (m)	Temperature (°C)	*S.* Enteritidis ATCC 13076 (log N_0_ − N)	*S.* Schwarzengrund (log N_0_ − N)
1	0.36	17.5	17.5	1.33	1.9
2	1	5	5	1.28	1.6
3	1	30	5	2	4.17
4	1	5	30	1.5	1.95
5	1	30	30	1.97	3.74
6	3	17.5	3.52	3.15	4.99
7	3	17.5	38.52	1.39	3.49
8	3	3.52	17.5	2.16	2.02
9	3	38.52	17.5	3.8	3.41
10	3	17.5	17.5	2.08	3.43
11	3	17.5	17.5	2.04	4.89
12	3	17.5	17.5	2.7	3.51
13	3	17.5	17.5	2.76	3.85
14	5	5	5	5.8 *	4.59
15	5	30	5	3.59	7.28 *
16	5	5	30	4.05	6.1 *
17	5	30	30	5.19	5.1
18	6.36	17.5	17.5	7.52 *	8.96 *

* Data expressed in logarithmic scale.

**Table 2 antibiotics-13-00068-t002:** Validation of the experimental model using random points to compare observed and predicted values.

Points	Treatments			*Salmonella* Enteritidis ATCC 13076		*Salmonella* Schwarzengrund	
	Chlorine ppm	Time (min)	Temperature (°C)	* Observed	* Predicted	* Observed	* Predicted
1	3.5	2.5	17.5°	2.84	3.400821	4.24	4.08
2	2.5	8	17.5°	2.579784	2.397442	3.06694679	2.82
3	4	3.5	17.5°	3.826391	4.036951	3.903089987	3.74
4	1.5	25	17.5°	2.253022	1.983209	3.678766618	3.17
5	3	19	17.5°	3.051153	2.641538	4.293204658	3.98

* Observed and predicted expressed in Log_10_ CFU/mL. **Equation** *S.* Enteritidis ATCC 13076: Log CFU/mL = 2.1385149129786 − 0.47257539167557 × Chlorine + 0.24174445563888 × Chlorine^2^ + 0.014189033415319 × Time − 0.0092024974789091 × Chlorine × Time. **Equation**
*S.* Schwarzengrund: Log CFU/mL = 1.6274450806394 + 0.0042166941127924 × Chlorine + 0.14263741135157 × Chlorine^2^ + 0.13585792619391 × Time − 0.0024777863722644 × Time − 0.62839913

**Table 3 antibiotics-13-00068-t003:** Indices of CCRD model performance and survival of *Salmonella* Enteritidis ATCC 13076 and *Salmonella* Schwarzengrund under chlorine concentration, temperature, and exposure time.

Models	Normality of Data (*p* Value) *	Normality of Residuals (*p* Value) *	R^2^adj	MSE	B_f_	A_f_	Lack of Fit (*p* Value)
*S.* Enteritidis ATCC 13076	0.2971	0.1519	0.82337	0.1265	1.13	1.42	0.1100
*S.* Schwarzengrund	0.296	0.4504	0.83952	0.094	1.1	1.36	0.3956

* Normally tested using the Shapiro–Wilk test.

**Table 4 antibiotics-13-00068-t004:** Variables used in CCRD. Factorial design points (−1; +1), smallest and largest values used (−1.44; +1.44), and center point (0).

Factors	Variables		
	Chlorine (ppm)	Time (m)	Temperature (°C)
−1.44	0.36	3.52	3.52
−1	1	5	5
0	3	17.5	17.5
1	5	30	30
1.44	6.36	38.5	38.5

## Data Availability

Data are contained within the article and Appendix A.

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
