# Peer review of "Optimizing the Antimicrobial Activity of Sodium Hypochlorite (NaClO) over Exposure Time for the Control of Salmonella spp. In Vitro"

_antibiotics, 2024, doi:10.3390/antibiotics13010068_

Round 1
Reviewer 1 Report
Comments and Suggestions for Authors
Salmonella are major foodborne pathogen and of public health concern globally. In this study the effect of NaClO under different conditions to reduce the load of Salmonella in fish was focused. It’s a well designed study and data are nicely presented. I found this manuscript very interesting and can contribute to the treatment of Salmonella in fish using NaCLO.
Few comments:
In introduction, please add few statements on the impact of Salmonella in fish related foodborne illness
What was the number of bacteria used in each of these experiment in terms of CFU?? Is the number correlate with the real scenario/fish.
What was the basis of selecting those temperature and con. /ppm of NaClO for the treatment. I was expecting to see the result of in vivo test to see the effect of NaClO.
Discuss the potential health hazards linked with NaClO if any??
Line 208, will it be this or in previous study?? Check..
Line 230, The National Health Surveillance Agency (ANVISA) approves the use of sanitizers 230 at a concentration of 5 ppm, as this concentration is sufficient to ….
Is this 5 ppm is also applicable for NaCLO ??
Author Response
Antibiotics - Manuscript ID antibiotics-2788523
Reviewer 1:
Comments to the Author:
Salmonella are major foodborne pathogen and of public health concern globally. In this study the effect of NaClO under different conditions to reduce the load of Salmonella in fish was focused. It’s a well designed study and data are nicely presented. I found this manuscript very interesting and can contribute to the treatment of Salmonella in fish using NaCLO.
Response: Dear Reviewer, thank you for your insightful comments and valuable suggestions regarding the structure of our manuscript. We have carefully considered your feedback and made the following revision:
Comments
In the introduction, please add a few statements on the impact of Salmonella in fish-related foodborne illness.
Response: We appreciate your suggestion on this matter. Accordingly, we have expanded the introduction to include information about the impact of Salmonella in fish-related foodborne illnesses, specifically in paragraph 5 of the revised manuscript.
What was the number of bacteria used in each of these experiment in terms of CFU?? Is the number correlate with the real scenario/fish.
Response: Approximately 7 log10 CFU/ml was used as the initial bacterial load for each Central Composite Rotatable Design (CCRD) experiment. This is specified in the manuscript as 'Quantification of the initial bacterial load was carried out with an inoculum at 7 log10 CFU/ml', as noted in Line 331. The lack of correlation between the numbers and the actual situation or fish species arises from the absence of definitive reports in existing literature regarding contamination in real fish. This gap highlights the urgent need for further research. It particularly points out the absence of a quantitative standard for native fish in Brazil, underlining the necessity for more comprehensive studies in this area.
What was the basis of selecting those temperature and con. /ppm of NaClO for the treatment. I was expecting to see the result of in vivo test to see the effect of NaClO.
Response: Thank you for your question. Central Composite Rotatable Design (CCRD) is a collection of statistical and mathematical techniques used to investigate the interrelationships between one or more responses and multiple factors in a study. In our research we identify the variables to be studied and establish random points for these variables. The study explored a range of chlorine concentrations, starting from the standard recommendation in Brazil (5 ppm) and extending to both higher and lower levels. This variation was examined in relation to different temperatures. Notably, no specific guidelines exist in Brazil regarding temperature adjustments for chlorine treatment, prompting our investigation into various temperature ranges. These included lower temperatures, ambient conditions (particularly referencing the climate of Mato Grosso), and warmer temperatures. Additionally, our research will be complemented by in vivo tests. We plan to leverage previous studies conducted by our team, aiming to test and validate the findings of this current research, thereby ensuring a comprehensive and robust analysis.
Discuss the potential health hazards linked with NaClO if any??
Response: Thank you for your suggestion. We discuss this subject in paragraph 5 of the study, line 242 to 249.
Line 208, will it be this or in previous study?? Check.
Response: The point in question refers to findings from a previous study.
Line 230, The National Health Surveillance Agency (ANVISA) approves the use of sanitizers at a concentration of 5 ppm, as this concentration is sufficient to ….
Is this 5 ppm is also applicable for NaCLO??
Response: Thank you for your observation, NaClO is approved by ANVISA at a concentration of 5 ppm as it is the only sanitizer that can have direct contact with the fish during its processing. We have added this information throughout the text on Line 234 and 235.
Reviewer 2 Report
Comments and Suggestions for Authors
Very nice paper, well explained, you can find my comments in the pdf I am attaching.

Comments on the Quality of English LanguageWell written and understandable.
Author Response
Antibiotics - Manuscript ID antibiotics-2788523
Reviewer 2:
Comments to the Author:
Very nice paper, well explained, you can find my comments in the pdf I am attaching.
Response: Dear Reviewer, thank you for your constructive feedback and positive comments. We have carefully addressed the details of the revision as per your suggestions and have incorporated these changes into the manuscript.
Points to observe:
Line 23, I suggest to refer in the text "viable population".
Response: Thank you for your consideration. The modification was made according to your suggestion.
Line 25, "Different time frames"-sounds maybe more appropriate.
Response: Thank you. The modifications were made to the Line 25.
Line 29, "inhibit the growth of the mentioned bacteria..."
Response: Thank you for your suggestion. The modification was made in Line 30.
Line 32, "that by...or... that with the ".
Response: Thank you. The modifications were made to the Line 33.
Line 45, "production". Producers is not, in my opinion, the most adequate word to use in this context.
Response: The modification was made according to your suggestion.
Line 65, "of the..."
Response: The modification was made in Line 65.
Line 111, "it is".
Response: The modifications were made to the Line 119.
Line 116, "that by..."
Response: The modifications were made to the Line 123.
Line 205, should be "log" in small letters.
Response: Thank you. Modifications were made throughout the text.
Line 342, shouldn't it be "BS"?
Response: You are correct, and we appreciate your keen observation. The necessary correction has been made to 'BS' in Line 356 of the manuscript. Thank you for bringing this to our attention.
Line 371, please rewrite this last sentence. “However, it is worth noting that the present study has not yet been standardized for in vivo use, and further research can be conducted using the time/concentration combination of NaClO in fish to reduce contamination.”
Response: Thank you for your suggestion. The modification was made to Line 385 to 388.
Reviewer 3 Report
Comments and Suggestions for Authors
This is an interesting work in the field of food microbiology, that does not fit perfectly into the scope of the journal Antibiotics. Apart from this, the main drawback of this study is the fact that all experiments were conducted in vitro; nothing was done in situ. Thus, the title "Optimizing the antimicrobial activity of sodium hypochlorite (NaClO) over exposure time for the control of Salmonella spp. in native fish" is not adequate. With such a title, additional experiments performed in situ on the selected fish carcass (or some specific fish parts) are necessary. However, the authors investigated the effect of selected NaOCl concentrations applied during different exposure times and at different temperatures, on the viability of selected Salmonella strains grown in vitro and resuspended in saline. Such an experimental design seems inappropriate and without enough novelty.
In addition to the above mentioned, some minor changes labeled directly in the text are needed.
With the additional experiments' results and major changes, the manuscript could be reconsider for the publishing.

Comments on the Quality of English LanguageSome editing of English language is required.
Author Response
Antibiotics - Manuscript ID antibiotics-2788523
Reviewer 3:
Comments to the Author:
This is an interesting work in the field of food microbiology, that does not fit perfectly into the scope of the journal Antibiotics. Apart from this, the main drawback of this study is the fact that all experiments were conducted in vitro; nothing was done in situ. Thus, the title "Optimizing the antimicrobial activity of sodium hypochlorite (NaClO) over exposure time for the control of Salmonella spp. in native fish" is not adequate. With such a title, additional experiments performed in situ on the selected fish carcass (or some specific fish parts) are necessary. However, the authors investigated the effect of selected NaOCl concentrations applied during different exposure times and at different temperatures, on the viability of selected Salmonella strains grown in vitro and resuspended in saline. Such an experimental design seems inappropriate and without enough novelty.
In addition to the above mentioned, some minor changes labeled directly in the text are needed.
With the additional experiments' results and major changes, the manuscript could be reconsider for the publishing.
Response: Dear Reviewer, thank you for your insightful comments. After careful consideration of your feedback and that of other reviewers, we agree that revising our title would be appropriate. This decision is based on the fact that we did not conduct in vivo experiments in this study. Our primary aim was to test and evaluate the results in experimental settings with fish to ascertain any significant effects. However, as this was not executed in the current study, we acknowledge the need for a title adjustment to more accurately reflect the scope of our research.
We deeply appreciate your comments and suggestions regarding the structure of our manuscript. We have meticulously addressed these in the following revisions:
Points to observe:
Line 72, ...interventions. All these factors can lead...
Response: Thank you for suggesting. The modification was made to Line 81.
Line 78-79, ...and concerning the efficiency of the forms of its control, it is still not well known.
Response: The modifications were made to the Line 87-88.
Line 81-82-83-84, The sentence should be revised, it is not written correctly.
Response: Thank you for pointing out the issue with the sentence spanning Lines 81-84. We appreciate your keen observation. Following your suggestion, we have thoroughly revised and corrected the sentence, now located in Lines 90-91-92-93 of the manuscript.
Line 94, The sentence is unclear and has to be revised.
Response: Thank you for the suggestion. The modifications were made to the lines 102-105.
Line 120, Why were only points 4, 5 and 1 presented. All points in the range 1-5 are featured with indicated values of temperature, time and NaOCl concentration.
Response: Thank you for your insightful observation regarding Line 120. Upon review, we realized the omission and have now included all points from 1 to 5, along with their corresponding values of temperature, time, and NaOCl concentration, in the revised manuscript at Line 128.
Line 134, However, the exposure time was not the same. It is written in the Table 1 that the exposure time was 30 min in the point 15, and 5 min in point 16.
Response: We made modifications to the line 142 according to your suggestion.
Line 138, bacterial reduction was not achieved or the treatment was ineffective.
Response: The modifications were made to the Line 147.
Line 150, Validation of the inactivation model of....
Response: Thank you for suggesting. The modification was made to Line 159.
Line 155, It is necessary to denote if the presented values were the log10 CFU/ml or Δ Log CFU/mL, it is now nuclear.
Response: Thank you for the observation, we added the requested comment to the line 164-165.
Line 216, The authors should clearly state why did they screen out this concentration activity if it is unfeasible. “Therefore, the concentration of 6.36 ppm used in our study is unfeasible for industries, as it is exceeds the recommended level”.
Response: Thank you for pointing out the need for clarification regarding the choice of concentration in Line 216. We chose to use a concentration of 6.36 ppm, which is higher than the recommended level, to test the hypothesis that a higher concentration results in greater inactivation of the tested strains. This concentration was selected specifically for experimental purposes. We acknowledge that it exceeds the industry-recommended limits, making it impractical for industrial use. We have elaborated on this rationale and included it in the revised manuscript at Lines 225-226 and 227.
Line 227, The sentence has to be revised, it is grammatically incorrect. “Therefore, we tested different concentrations at various times and temperatures to opti-mize this the use of this substance in refrigerated slaughterhouses”.
Response: The modifications were made to the Line 237-238 and 239.
Line 236, The sentence has to be revised, it is grammatically incorrect and unclear. “Adaptations to sanitizers can and should be avoided, so that pathogenic bacteria do not create resistance, not using them below recommended concentration”.
Response: The modifications were made to the Line 243-244.
Line 237, The sentence has to be revised, it is grammatically incorrect. “Prudent use is essential their continued effectiveness against microorganisms”.
Response: Thank you for observation. The modifications were made to the Line 244-245.
Line 256, This is unclear and should be revised. Why could the correct fish washing with NaClO solution induce damage in the case of fish contamination with Salmonella spp?
Response: We appreciate your keen observation regarding the incorrect information. Following your suggestion, we have removed this erroneous detail from the manuscript. Thank you for helping us improve the accuracy of our work.
Line 273, unclear, should be revised. “The lack of fit provides an estimate of the fit of the experimental model”.
Response: Thank you for pointing out the lack of clarity in Line 273. We have revised the sentence to better convey the purpose of the lack of fit test. It now reads: 'The lack of fit test provides an estimate of how well the experimental model fits the data, determining whether the model is adequate. In the present study, this was found to be not significant for the models, with a p-value greater than 0.05.' These modifications have been made to Lines 285-288 in the revised manuscript.
Line 276, submitted to selected sodium hypochlorite concentrations during variable exposure time and on different temperatures.
Response: Thank you for the suggestion, we added the requested comment to the lines 290 and 291.
Reviewer 4 Report
Comments and Suggestions for Authors
The manuscript entitled “Optimizing the antimicrobial activity of sodium hypochlorite (NaClO) over exposure time for the control of Salmonella spp. in native fish” by Nunes et al. is scientifically valuable, well designed and conducted, with original and relevant contributions on identifying the most effective strategies to reduce Salmonella contamination in the fish industry. However, this study requires minor corrections and improvements, which will allow the publication in the prestigious journal Antibiotics.
The table numbering is incorrect. Also, the citation of tables in the main manuscript is not correct. Please correct these errors, which create difficulties in understanding the study and the results obtained.
In Table 4, please replace the concentration “0,36” with the correct form “0.36”.
The list of references should be reviewed and be prepared according to the requirements of the journal.
The tables included in Appendix A are untitled and have no explanatory notes. Please check and correct.
I suggest that these tables be uploaded as supplementary material.
Author Response
Antibiotics - Manuscript ID antibiotics-2788523
Reviewer 4:
Comments to the Author:
The manuscript entitled “Optimizing the antimicrobial activity of sodium hypochlorite (NaClO) over exposure time for the control of Salmonella spp. in native fish” by Nunes et al. is scientifically valuable, well designed and conducted, with original and relevant contributions on identifying the most effective strategies to reduce Salmonella contamination in the fish industry. However, this study requires minor corrections and improvements, which will allow the publication in the prestigious journal Antibiotics.
Response: Dear Reviewer, we are grateful for your positive evaluation of our manuscript. We appreciate your suggestions for minor corrections and improvements. We have diligently addressed each of these points and provided detailed revisions as per your recommendations.
Points to observe:
The table numbering is incorrect. Also, the citation of tables in the main manuscript is not correct. Please correct these errors, which create difficulties in understanding the study and the results obtained.
Response: Thank you for bringing this to our attention. We acknowledge the issue with the incorrect numbering of tables and the improper citation of these tables in the main manuscript. We have carefully reviewed and corrected these errors to ensure that the tables are correctly numbered and accurately cited.
In Table 4, please replace the concentration “0,36” with the correct form “0.36”.
Response: We agree with your observation and have corrected the concentration in Table 4 from '0,36' to the appropriate form '0.36'. Thank you for pointing out this error.
The list of references should be reviewed and be prepared according to the requirements of the jornal.
Response: Thank you for pointing out the issues with our list of references. We have thoroughly reviewed and revised the references to ensure they align with the journal's requirements. The necessary corrections and formatting adjustments have been meticulously made throughout the reference section.
The tables included in Appendix A are untitled and have no explanatory notes. Please check and correct. I suggest that these tables be uploaded as supplementary material.
Response: Thank you for your suggestion regarding the tables in Appendix A. We have now added titles and explanatory notes to each table for better clarity and understanding. Additionally, following your recommendation, these tables have been prepared to be uploaded as supplementary material to complement the main manuscript.
Round 2
Reviewer 3 Report
Comments and Suggestions for Authors
The authors have made the necessary improvements of the manuscript. It is now acceptable for publishing.